# Understanding Passive Membrane Permeation of Peptides: Physical Models and Sampling Methods Compared

**DOI:** 10.3390/ijms24055021

**Published:** 2023-03-06

**Authors:** Liuba Mazzanti, Tâp Ha-Duong

**Affiliations:** BioCIS, CNRS, Université Paris-Saclay, 17 Avenue des Sciences, 91400 Orsay, France

**Keywords:** peptide membrane permeability, molecular dynamics simulation, umbrella sampling, Markov State Model, free energy profile

## Abstract

The early characterization of drug membrane permeability is an important step in pharmaceutical developments to limit possible late failures in preclinical studies. This is particularly crucial for therapeutic peptides whose size generally prevents them from passively entering cells. However, a sequence-structure-dynamics-permeability relationship for peptides still needs further insight to help efficient therapeutic peptide design. In this perspective, we conducted here a computational study for estimating the permeability coefficient of a benchmark peptide by considering and comparing two different physical models: on the one hand, the inhomogeneous solubility–diffusion model, which requires umbrella–sampling simulations, and on the other hand, a chemical kinetics model which necessitates multiple unconstrained simulations. Notably, we assessed the accuracy of the two approaches in relation to their computational cost.

## 1. Introduction

To reach an intracellular target, a drug must cross the cellular membrane. This process can be performed by endocytosis, which is generally involved in internalizing macromolecules or drug nanocarriers [1], or by using membrane transporter proteins, such as the human ATP binding cassette (ABC) [2] and solute carrier (SLC) [3] families, or the outer membrane porins [4] and TonB-dependent transporters [5] of gram-negative bacteria. However, the predominant mechanism for xenobiotics is a passive diffusion across the lipid bilayer along a concentration gradient [6,7]. It is, thus, essential in pharmaceutical developments to characterize the drug membrane permeability to anticipate their pharmacokinetic properties. This is particularly crucial for peptide-based therapeutics. Indeed, peptides are very promising compounds for modulating protein-protein interactions (PPIs), especially those involved in intracellular signaling pathways [8]. This is partly due to their size, which is well-appropriate for covering the generally large molecular surfaces involved in PPIs. However, this size may turn out to be a disadvantage for membrane permeation.

Passive membrane permeability of drugs can be characterized experimentally by using the parallel artificial membrane permeability assay (PAMPA), which consists in measuring the number of compounds that have crossed a planar artificial membrane from donor to acceptor wells [9]. Alternative methods have been developed to monitor in real-time the entry of drugs into unilamellar liposomes using fluorescent probes [10,11]. In both approaches, the drug membrane permeability *P* is quantified as P=J/ΔC, where *J* is the flux of the drug through the membrane and ΔC the difference between the donor (or outside) and acceptor (or inside) drug concentrations.

Throughout a membrane permeation process, a peptide can bind the lipid headgroup-water interface, fold into secondary structures, form aggregates, insert into the lipid hydrophobic tails, and perturb the bilayer organization [12,13]. These multiple possible events depend on the peptide apolar/polar/charged amino acid composition, its structure and flexibility, the solvent pH, the ionic strength, and the membrane content in lipids. All these factors make it challenging to predict the membrane permeability and deeply understand the mechanism of peptide translocation. To tackle this issue, molecular dynamics (MD) simulations of peptide crossing lipid bilayers are valuable tools to gain insight into the physical-chemical factors that govern the permeation process.

However, membrane permeation occurs on a millisecond to second timescale [10] which are generally out of reach for classical MD simulations. Thus, enhanced sampling methods, such as metadynamics or umbrella sampling (US), are needed to provides valuable information about the compound propensity to partition between water and membrane, the structures of the intermediate states within the bilayer, and about the mechanisms of permeation [14,15]. In particular, trajectories from US simulations allow to compute drug free energy profiles (FEP) across lipid bilayers and can be employed to estimate their membrane permeability by using the so-called inhomogeneous solubility-diffusion model (ISDM) proposed by Marrink and Berendsen [16]. This approach has been applied to several drugs over the past few years [14,17,18,19] and recently to peptides [20].

Since membrane permeability is a kinetic property, it can be naturally computed by using a kinetic model of the permeation process. This consists in describing the elementary steps of a compound membrane permeation and formulating each of the corresponding reaction rates as a function of the concentrations in the intermediate states. Then, the master differential equations are solved to yield the compound concentrations on both sides of the membrane as a function of time, from which the membrane permeability can be retrieved with Fick’s law of diffusion [18]. Importantly, this method requires to preliminary determine the rate constants of the kinetic model. This can be done with Markov State Model (MSM) analyses which can extract long-time kinetic information of a molecular system from an ensemble of “short” MD trajectories starting from multiple initial configurations [21,22].

For small compounds, calculations based on kinetic models appear to provide membrane permeability values that better correlate with experiments than those based on ISDM [18]. Nevertheless, as far as we know, the former approach has never been applied to peptides. In the present study, we comparatively assessed the two computational methods to estimate the membrane permeability for a benchmark cyclic peptide of 10 amino acid residues. Notably, we scrutinized the accuracy of the two approaches in relation to their computational cost.

## 2. Results and Discussion

We applied the two computational approaches to a paradigmatic ten amino acids cyclic peptide for which PAMPA assays were performed [9]. Constrained or unconstrained MD trajectories of the peptide assembled with a planar lipid bilayer were collected to ultimately calculate the peptide permeability coefficient through Equation (Equation 4) or (Equation 8) for the ISDM or the MSM-based model, respectively. In both approaches, the free energy profile of the peptide perpendicularly crossing the planar bilayer membrane is crucial for estimating its permeability coefficient. In the ISDM method, the latter is directly related to the FEP ΔG(z) through Equations (Equation 2)–(Equation 4). On the other hand, in the MSM-based approach, without the knowledge of the energy barrier location, the Markov State Models could be biased since the phase space sampling by short trajectories depends on the starting points along the collective variable. Therefore, the initial configurations must correspond to maximum energy states for the trajectories not to be trapped in energy minima. Thus, we will present first the free energy analyses and, subsequently, the quantitative estimates for the permeability coefficients.

### 2.1. Free Energy Profiles

The weighted histogram analysis method (WHAM) [23] applied to the umbrella sampling trajectories for the cyclic decapeptide designated as CDP5 (Section 3.1) yields the free energy profile displayed in Figure 1. Moving from the bulk water to the membrane, the peptide encounters a first energy barrier of about $\Delta G_{in}$ = 3 kcal/mol located on the surface of the lipid headgroups in the water phase (z=2.4 nm). Then the FEP exhibits a minimum on the other side of the lipid headgroup plane, inside the membrane (*z* = 1.4 nm), where the amphiphilic peptide is stabilized by both hydrophobic and polar interactions with the lipid tails and headgroups, respectively. We note that the free energy of this stable state is roughly equal to that one in the bulk water. Therefore, the free energy barrier for moving out from the interior of the membrane into the bulk water is also $\Delta G_{out}=\Delta G_{in}=3$ kcal/mol. The major energy barrier for crossing the membrane is located at its center (*z* = 0.0 nm), indicating that the rate-limiting step during the peptide permeation is the flip-flop passage from one lipid layer to the other. The necessary energy to overcome this barrier is about $\Delta G_{flip}$ = 12 kcal/mol.

The FEP computed for the CDP5 peptide is very similar to those of amphiphilic cyclic hexapeptides reported by Sugita et al. [20], indicating that these cyclic peptides of comparable size should have the same mechanism of membrane translocation in which the flip-flop event within the bilayer is the rate-limiting step. This contrasts with the FEPs of amphiphilic small compounds computed by Dickson et al. [18], which exhibit two marked energy minima located at the inner lipid headgroup surfaces much lower than the free energy in the bulk water. For these small compounds, the energy barrier at the membrane center is generally small, depending on their hydrophobicity, indicating that the major rate-limiting step of their permeation process is moving from the interior of the membrane into the bulk water.

To compute the peptide membrane permeability using kinetic models, we built a first Markov State Model (*CDP5_2.4*) from unconstrained MD simulations of the peptide with initial positions at *z* = 2.4 nm and *z* = 0.0 nm corresponding to the two maxima of its FEP computed from US trajectories. Moreover, to assess the sensitivity of this approach to the initial configurations, we also built Markov State Models (*CDP5_2.3* and *CDP5_2.6*), respectively from trajectories starting at *z* = 2.3 nm and *z* = 0.0 nm on the one hand, and from trajectories starting from *z* = 2.6 nm and *z* = 0.0 nm, on the other hand. A fourth Markov State Model (*CDP5_all*) was also built using all the aforementioned trajectories (Section 3.3). From each of these sets of unconstrained MD trajectories, we computed the peptide stationary distributions π and the free energy profiles $\Delta G=-k_{B}T\log\pi$, which are shown in Figure 2.

FEPs derived from unconstrained trajectories show significant differences with respect to the profile computed from the US trajectories: First, the energy barriers to cross the lipid headgroups are smaller (by about 2 kcal/mol) than the US ones, except for the (*CDP5_2.6*) set of trajectories. Secondly, the positions of the energy minima inside the membrane are shifted toward the lipid tails (*z* = 1.0 nm) compared to the US FEP (*z* = 1.4 nm). However, the most striking difference is the free energy barrier corresponding to the flip-flop step: the unconstrained simulations yield a central free energy barrier of about 2 kcal/mol, while the US flip-flop barrier almost reaches 12 kcal/mol. This suggests that the peptide membrane permeability estimated with the kinetic models will be much higher than in the ISDM-based approach.

It is interesting to note that the four FEPs generated by the four sets of unconstrained simulations display different qualitative behaviors, implying that a slight change in their initial conditions could entail a sensible variation in the results. In particular, we notice that the free energy needed to translocate the peptide from inside the membrane into the water phase is larger than the energy barrier associated with the flip-flop step in the *CDP5_2.6* model, while it is smaller in the three other cases (Figure 2). The emergence of high in-out free energy barriers in the *CDP5_2.6* case is a consequence of the initial peptide positions, which were slightly shifted from the minor maxima of the US FEP (*z* = 2.4 nm) toward the water bulk. Therefore, fewer peptide trajectories could naturally sample this maximum energy state which increases the barrier height, compared to the three other cases (where it could even be observed a local shallow energy minimum at this location). This highlights the sensitivity of the peptide FEPs computed from unconstrained MD simulations upon their initial configurations.

### 2.2. Membrane Permeability Calculations

In the framework of the ISDM approach, an estimate of the peptide permeability coefficient can be directly obtained from the US trajectories by extracting the integrated autocorrelation times $\tau_{zz}$, the collective variable variance $\sigma_{z}^{2}$, and the free energy profile ΔG(z) and applying Equations (Equation 2)–(Equation 4). For the CDP5 peptide, we get the value reported in Table 1. Compared to experiments, the ISDM-based estimation of the peptide membrane permeability is about one order of magnitude lower than the value found in PAMPA assays [25].

On the other hand, further calculations and analysis are necessary to get an estimate of the permeability coefficient within the liposome kinetic model approach. Rate constants are derived as the inverse of the mean first passage times (MFPTs) and then converted to apparent rate constants for the liposome model using Equation (Equation 6). As detailed in Appendix A, integration of the kinetic model Equation (Equation 5) yields a three-exponential time evolution for the substrate in the inner aqueous compartment (Equation (Equation 15)). By substituting $k_{I}$ of Equation (Equation 8) with the fastest rate constants $\kappa_{n}$ appearing in the substrate time evolution, we obtained the permeability coefficients reported in Table 1. Compared to the PAMPA value, it appears that the liposome kinetic models overestimate the peptide membrane permeability by about five to six orders of magnitude.

While comparing theoretical with experimental estimates of the peptide membrane permeability, we should bear in mind that PAMPA assays use hexadecane plates separating two water compartments [25] instead of a lipid bilayer or spherical liposomes. Thus, if we assume that crossing a hexadecane plate is easier than a lipid bilayer, this difference in the membrane nature might explain the lower permeability found by the ISDM-based method compared to experiments. On the other hand, the permeability estimated with liposome kinetic models being proportional to the liposome radius, it should be intuitively much lower than the permeability through a planar membrane. This is the opposite tendency that we can notice in Table 1 and, altogether, it appears that the MSM-based approach greatly overestimates the peptide membrane permeability compared to experiments.

The discrepancy between the ISDM-based and the MSM-based approaches can be understood as a consequence of the difference in the free energy barriers of the FEPs computed by the two methods. As the energy barrier for the flip-flop step is larger in the US FEP than in the unconstrained simulations-derived one, permeation through the membrane is intuitively less easily achieved within the ISDM framework than from the standpoint of the MSM-based liposome kinetic models. To a lesser extent, the slight variations of the energy barrier between the MSM-based FEPs can account for the slight differences in the permeability coefficient estimations (Table 1).

## 3. Methods and Materials

### 3.1. Building the Peptide-Membrane System

The peptide that we chose for the benchmark of the two methods that compute membrane permeability is a cyclic decapeptide named CDP5 (Figure 3). Four out of its ten residues were N-methylated to improve its cell permeability and oral bioavailability [26,27]. Its membrane permeability coefficient was measured using PAMPA experiments [9] with a hexadecane layer separating the donor and acceptor compartments [25]. It should be noted that this hexadecane layer differs from the lipid bilayer that will be considered in our simulations and which is a more realistic model of experimental liposomes. Nevertheless, as far as this study is concerned, the experimental value of CDP5 permeability coefficient reported in [25] ($\log P_{exp}=-6.7$) will be used as a reference for our theoretical results.

The modified peptide CDP5 was parameterized using the highly optimized chemical building blocks from CGenFF [28] to be compatible with the CHARMM36 force field for lipids [29]. An initial three-dimensional structure was generated from its sequence using OpenBabel [30] and PyMol [31]. After an energy minimization of 5000 steps and two short equilibration runs (1 ns in NVT and 2.5 ns in NPT ensemble), CDP5 was submitted to a 100 ns MD simulation in water with a sodium chloride concentration of 0.15 mol/L, using a V-rescale thermostat and a Parrinello-Raman barostat set at 300 K and 1 bar, respectively. Lennard-Jones potentials were cut off at 1.2 nm, electrostatic interactions were calculated using the smooth PME method [32], and covalent bonds with hydrogen were constrained using LINCS algorithm [33], for all simulations in this work. The resulting trajectory showed strong structural stability of the CDP5 peptide and was clustered based on RMSD with a cutoff value of 0.15 nm. The representative structure of the most populated cluster (Figure 3) provided the peptide conformation to be assembled with the membrane.

A lipid bilayer composed of 25 POPC (1-palmitoyl-2-oleoyl-sn-glycero-3-phosphocholine) molecules per leaflet were built at the temperature of *T* = 293.15 K using CHARMM-GUI [34]. The membrane was energy minimized and equilibrated in water with 0.15 mol/L of NaCl by running a 250 ps NVT and a 125 ps NPT equilibration with a semi-isotropic pressure scaling, followed by three 500 ps NPT runs with decreasing position restraints on lipid phosphate groups from 1000 to 0 kJ/mol/Å${}^{2}$. Then, the peptide and the lipid bilayer were assembled in a rectangular box, solvated, energy minimized, and equilibrated with 1 ns NVT and 1 ns NPT runs by decreasing the position restraints on peptide backbone heavy atoms and lipid phosphate groups from 4000 and 1000 to 50 and 0 kJ/mol/Å${}^{2}$, respectively. The final conformation of the equilibration procedure was taken as the initial configuration for US simulations.

GROMACS 2021 [35] was used for all simulations, with CHARMM36 [29] force-field for lipids, and TIP3P [36] model for water.

### 3.2. ISDM-Based Calculation of Membrane Permeability

This subsection presents the calculations that were performed to estimate the peptide membrane permeability by using the inhomogeneous solubility-diffusion model (ISDM) [16].

Umbrella Sampling and WHAM Analysis

In the initial configuration of the peptide-membrane system, the solute is located in the bulk water at a distance $z_{max}$ = 4 nm from the center of the lipid bilayer (the membrane plane being oriented perpendicular to the z-axis). Starting from this initial position, the solute was pulled toward and inside the lipid membrane by an increment of Δz = 0.1 nm until it reached the position $z_{min}$ = 0 nm corresponding to the center of the bilayer. For each peptide position (or window), which is maintained with an umbrella potential of 1000 kJ/mol/${Å}^{2}$, energy minimization was performed, followed by a 1 ns NVT and 1 ns NPT equilibration. Then, a production simulation of 250 ns was performed. The final conformation of the equilibration stage was pulled and used as a starting conformation for the next window.

The last 130 ns of the production trajectories were analyzed with the WHAM analysis [23] implemented in GROMACS [24]. More specifically, taking as input the values of the collective variable *z* for each window, gmx wham tool was used to compute both the free energy profile $\Delta G\left(z\right)=G\left(z\right)-G\left(z_{max}\right)$ and the normalized integrated autocorrelation time $\tau_{zz}$ defined as:(1)$$\tau_{zz}=\frac{1}{\sigma_{z}^{2}}\int_{0}^{\infty }\left[z(0)-\langle z\rangle \right]\left[z(t)-\langle z\rangle \right]dt$$
where $\sigma_{z}^{2}$ is the variance of the collective variable *z* defined as $\sigma_{z}^{2}=\langle z^{2}\rangle -{\langle z\rangle }^{2}$. It could be noted that gmx wham provides both averages and standard errors of the FEP and $\tau_{zz}$ by using bootstrap calculations [24].

Inhomogeneous Solubility-Diffusion Model

The ISDM considers that, due to the inhomogeneous nature of lipid membranes, the diffusion rate of a solute strongly depends on its position in the latter [16]. Moreover, it assumes that the diffusion process obeys the classical diffusion equation, implying that the dependence of $\langle z^{2}\rangle$ on time is linear. Thus, the position-dependent diffusion coefficient can be calculated as the ratio of the variance $\sigma_{z}^{2}$ over the integrated autocorrelation time $\tau_{zz}$ provided by WHAM analyses of US simulations:(2)$$D\left(z\right)=\frac{\sigma_{z}^{2}}{\tau_{zz}}$$

Using the diffusion coefficient D(z) and the free energy ΔG(z), the local resistance of the solute permeation through the membrane can be defined as [16]:(3)$$R\left(z\right)=\frac{\exp\left[\beta \Delta G(z)\right]}{D(z)}$$
where $\beta =1/(k_{B}T)$, with T the temperature and $k_{B}$ the Boltzmann’s constant. The sum of all the R(z) along the variable *z* gives the global resistance and its inverse yields the membrane permeability:(4)$$P_{ISDM}={\left[\int_{-z_{max}}^{z_{max}}R(z)dz\right]}^{-1}$$

To estimate the errors on the permeability coefficient, 100 bootstrapped trajectories were generated from the original trajectory of each window using an in–house script. Then, for each bootstrapped trajectory, the permeability coefficient was calculated as described above. The final results were computed as the mean of these 100 values and errors were given by the standard deviation.

### 3.3. MSM-Based Calculation of Membrane Permeability

In this subsection, we present the different steps of the method for computing the peptide membrane permeability by using a kinetic Markov State Model (MSM) [21].

Multiple Unconstrained MD Simulations

A kinetic Markov State Model is essentially based on the transition probabilities between the model discretized states. These probabilities can be estimated by running multiple unbiased MD simulations, which make the system pass through its different minimum energy states. Nevertheless, to sufficiently sample the transition events, it is recommended to run these simulations from out-of-equilibrium initial configurations, which can “go down the hill” to the nearest metastable states in a short time [21,22]. The identification of these maximum energy states implies the preliminary determination of the system’s free energy landscape. Regarding the permeation of CDP5 peptide, its FEP exhibits two main transition states, a major one at the center of the bilayer (*z* = 0 nm) and a minor one at the surface of the lipid headgroups (around *z* = 2.4 nm) (Figure 1). Consequently, we decided to run simulations of the peptide-membrane system, starting from four sets of 75 conformations each, extracted from the previous US simulations, with peptide positions *z* = 0.0, 2.3, 2.4, and 2.6 nm. Each of these 4 × 75 initial configurations were submitted to a 100 ns unconstrained MD production in the same conditions as the US simulations.

Building the Markov State Models

To study the sensitivity of the computed permeability to the initial configurations of the peptide-membrane system, we built four Markov models from four different sets of MD trajectories (Table 2).

PyEMMA [22] was used to build and analyze the four MSMs. For each kinetic model, the estimated implied relaxation timescale was plotted as a function of different lag times τ (Figure 4). From these graphs, we considered that the implied relaxation timescales are approximately constant from 3 timesteps, i.e., τ = 300 ps, and that this lag time is suitable for calculating accurate kinetic coefficients of all MSMs. These plots also indicate that the models are characterized by three implied slow relaxation timescales (blue, red, and green lines in Figure 4). The transition matrix was estimated using the above lag time value, and a Chapman–Kolmogorov test was performed to verify the Markovianity of each model [22].

Next, PCCA+ [37] clustering was applied to describe all trajectories of each MSM in terms of a few metastable states. Since three implied slow relaxation timescales could be identified, we asked PCCA+ to cluster the trajectories into four metastable states (Figure 5). Moreover, PCCA+ provides the values of the MFPTs between each pair of metastable states (Figure 5), which were used to derive the rate constants corresponding to the entry of the solute into the membrane $k_{in}$, the exit out of it $k_{out}$, and the flip-flop transitions from one lipid layer to the other $k_{flip}$ (Figure 6A).

Kinetic Model Master Equations

Similarly to what was done by Dickson et al. [18], the master equations describing the passive diffusion of a solute from an outer water compartment into the water interior of a spherical liposome were solved by considering two lipid compartments (indicated as outer and inner lipid compartments) as schematically depicted in Figure 6B. The differential equations describing the time dependence of the solute in the four compartments read:(5)$$\begin{array}{cc} \frac{dS_{I}\left(t\right)}{dt}= & \sum_{J}\left[-k_{IJ}S_{I}\left(t\right)+k_{JI}S_{J}\left(t\right)\right]\\ S_{wo}\left(0\right)=S_{tot}, & S_{lo}\left(0\right)=S_{li}\left(0\right)=S_{wi}\left(0\right)=0 \end{array}$$
where I,J=wo,lo,li,wi are the indices that refer to the outer water, outer lipid, inner lipid, and inner water compartments, respectively, and the summation runs over the nearest compartments. At time t=0, the solute is entirely in the outer water compartment (Equation (Equation 5)).

Knowing that the rate constant *k* of each kinetic step depends on the permeability coefficient *P*, the surface area *A* separating two compartments, and the volume *V* of the donor compartment, through the relation k=P(A/V), it should be noted that the apparent rate constants $k_{wo,lo}$, $k_{lo,wo}$, $k_{lo,li}$, $k_{li,lo}$, $k_{li,wi}$, and $k_{wi,li}$ in the membrane spherical geometry are related to the rate constants $k_{in}$, $k_{out}$, and $k_{flip}$ previously estimated by the Markov State Models in the membrane planar geometry as
(6)$$\begin{array}{ccccccc} k_{wo,lo} & = & \frac{V_{W}}{V_{wo}}\frac{A_{l}}{A_{L}}k_{in} & , & k_{wi,li} & = & \frac{V_{W}}{V_{wi}}\frac{A_{l}}{A_{L}}k_{in}\\ k_{lo,wo} & = & \frac{V_{L}}{V_{l}}\frac{A_{l}}{A_{L}}k_{out} & , & k_{li,wi} & = & \frac{V_{L}}{V_{l}}\frac{A_{l}}{A_{L}}k_{out}\\ k_{lo,li} & = & \frac{V_{L}}{V_{l}}\frac{A_{l}}{A_{L}}k_{flip} & , & k_{li,lo} & = & \frac{V_{L}}{V_{l}}\frac{A_{l}}{A_{L}}k_{flip} \end{array}$$
where $A_{L}$, $V_{L}$, and $V_{W}$ are the lipid bilayer surface area, its interior volume, and the bulk water volume in the planar membrane systems simulated by MD, respectively. In the spherical liposome model, the surface area is considered equal on both outer and inner layers: $A_{l}\equiv A_{li}=A_{lo}$. Similarly, we assumed that the volumes of the two compartments in the lipid bilayer are equal: $V_{l}\equiv V_{li}=V_{lo}$. Quantitatively, we used a liposome radius *r* = 100 nm and a ratio of 1000:1:1:10 for the volumes of the four compartments $V_{wo}:V_{lo}:V_{li}:V_{wi}$, conforming to the experimental conditions of the liposome experiments reported in [10]. For the MD areas and volumes, we used the POPC area per lipid $A_{L}=65.6$ Å${}^{2}$ and the volume per lipid $V_{L}=1.2$ nm${}^{3}$ [18]. The bulk water volume $V_{W}$ was calculated from the volume per molecule 30.5 Å${}^{3}$ and the number of water per lipid (89 in our simulated systems).

Equations (Equation 5) are a system of four ordinary differential equations (three of which are independent) that we solved numerically using SciPy routines [38]. We also solved the system analytically, using the numerical values of the eigenvalues of the matrix associated with the system of differential equations (Appendix A).

Permeability Coefficient from Kinetic Models

Following reference [39] and subsequent papers [10,40], the permeation coefficient is extracted from experimental data by fitting the luminescence curve with a biexponential function as a function of time:(7)$$I\left(t\right)=I\left(\infty \right)+\left[I{\left(\infty \right)}_{I}-I{\left(0\right)}_{I}\right]e^{-k_{I}t}+\left[I{\left(\infty \right)}_{II}-I{\left(0\right)}_{II}\right]e^{-k_{II}t}$$
where $I\left(\infty \right)=I{\left(\infty \right)}_{I}+I{\left(\infty \right)}_{II}$ is the total maximal luminescence, $I{\left(\infty \right)}_{I}$ and $I{\left(\infty \right)}_{II}$ are the maximal luminescence of the fast and slow phase respectively, $I{\left(0\right)}_{I}$ and $I{\left(0\right)}_{II}$ are the initial luminescence for the two phases, and $k_{I}$ and $k_{II}$ are the corresponding rate constants.

For luminescence experiments with liposomes, the permeation coefficient has been related to the rate constant $k_{I}$ of the fast exponential phase and the liposome radius as:(8)$$P_{lipo}=\frac{V_{wi}}{A_{l}}k_{I}=\frac{r}{3}k_{I}$$

From the point of view of the four-step model describing the process, where a peptide undergoes adsorption, flip-flop, and desorption through the liposome membrane bilayer, the relevant kinetics are expressed as three exponential functions, as in Equation (Equation 15), rather than the two fitting functions for the luminescence intensity of Equation (Equation 7). We can, however, still use the rate constant associated with the fastest exponential phase in the liposome model and substitute its value in Equation (Equation 8) to obtain the membrane permeability coefficient, as it has been done in [18].

Errors on the permeability coefficient are derived using the Bayesian estimator in PyEMMA for the MSMs, which generates samples of transition matrices rather than one single transition matrix, as in the case of the maximum likelihood estimator. With this method, we obtained a sample of 100 transition matrices for each MSM, from which PyEMMA can compute samples for different quantities, such as the MFPT. The reported results were evaluated based on the mean value of the MFPT samples and their standard deviation, using propagation of uncertainties to compute the errors on the rate constants.

## 4. Conclusions

The ISDM approach is probably the most popular method used to estimate the membrane permeability of chemical compounds. However, Dickson et al. recently showed that an alternative method, based on Markov State Models of the permeation process, yielded permeability coefficients in better agreement with experimental data for seven small molecules [18]. In the present study, we applied and compared the two computational methods for a larger compound, the cyclic decapeptide CDP5. As opposed to the tendency reported by Dickson et al., we found that the ISDM-based and MSM-based method slightly underestimates and largely overestimates the peptide membrane permeability, respectively.

A probable explanation of the peptide permeability overestimation by our MSM-based calculations lies in the too-short duration of its unconstrained MD simulations required to build the Markov State Models. Indeed, due to its size and polarity, the CDP5 decapeptide cannot pass through the lipid headgroups as easily and rapidly as the small compounds simulated by Dickson et al. [18]. Thus, the peptide unconstrained simulations starting from maximum energy positions on the lipid headgroup surface in water require longer times to go down the hill and move towards the energy minimum inside the membrane. As a consequence, the peptide population π and the free energy $\Delta G=-k_{B}T\log\pi$ at these maximum energy positions are larger and lower than expected, respectively. Accordingly, the MSM-based peptide permeability coefficient is excessively large compared to experiments and ISDM calculations.

To remedy this default, it would probably be necessary to extend the duration of the unconstrained MD simulations and/or multiply their starting points. However, for the building of one Markov State Model, we have already performed a total of 15 μs MD simulations, whereas the ISDM approach has needed 10 μs US. In addition, the building of accurate MSMs requires running unconstrained MD simulations from maximum free energy states, which, very often, have to be predetermined by US calculations. Thus, predictions of the membrane permeability of peptides with the MSM-based method would require much more computer resources than the ISDM approach, which was able to yield reasonable estimations.

## Figures and Tables

**Figure 1 ijms-24-05021-f001:**
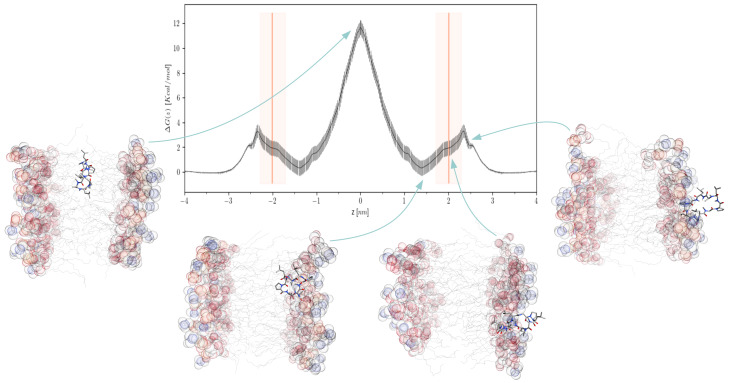
Symmetrized CDP5 free energy profile computed from umbrella sampling trajectories. Error bars were estimated using the gmx wham bootstrap algorithm [24]. The orange vertical line and shaded area indicate the mean and one σ of the lipid headgroup positions, respectively. Free energy difference $\Delta G\left(z\right)=G\left(z\right)-G\left(z_{max}\right)$ was calculated with $z_{max}$ = 3.8 nm in the water phase. Four representative structures of the peptide-membrane system are shown for the peptide positions *z* = 0.0, 1.4, 2.0, and 2.7 nm. Peptides, lipid headgroups, and lipid tails are displayed using sticks, semi-transparent spheres, and lines, respectively.

**Figure 2 ijms-24-05021-f002:**
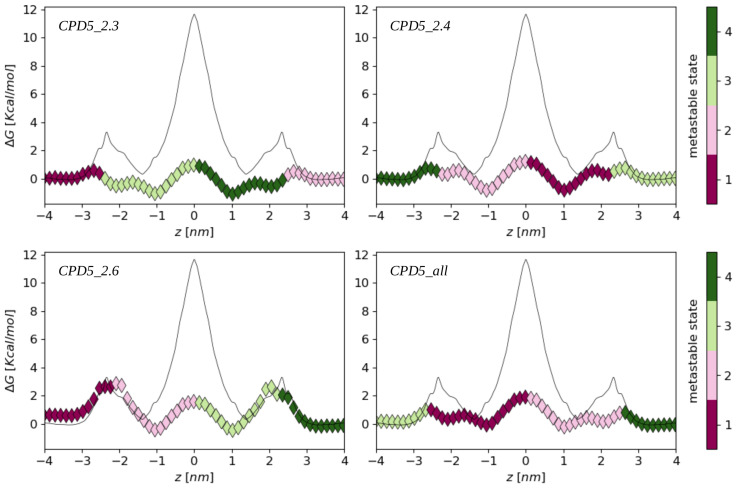
CDP5 free energy profiles computed from four different sets of unconstrained trajectories (Section 3.3). Diamonds indicating the discrete states are colored according to the metastable state to which they belong. FEPs were set to zero in the water phase, including the FEP computed from US trajectories (black lines).

**Figure 3 ijms-24-05021-f003:**
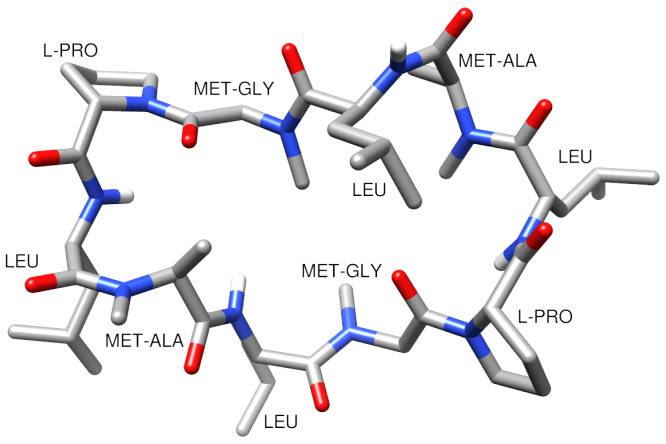
Sequence and structure of the cyclic decapeptide CDP5 with four N-methylated residues.

**Figure 4 ijms-24-05021-f004:**
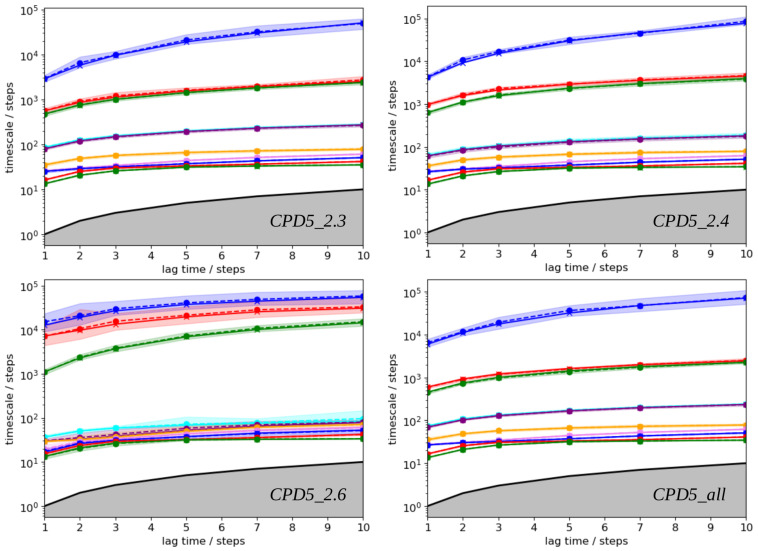
PyEMMA implied timescales as a function of lag time for each Markov State Model. Timescales are sorted in descending order using the color lines blue, red, green, cyan, violet, yellow, and pink. Black lines indicate timescale equal to lag time.

**Figure 5 ijms-24-05021-f005:**
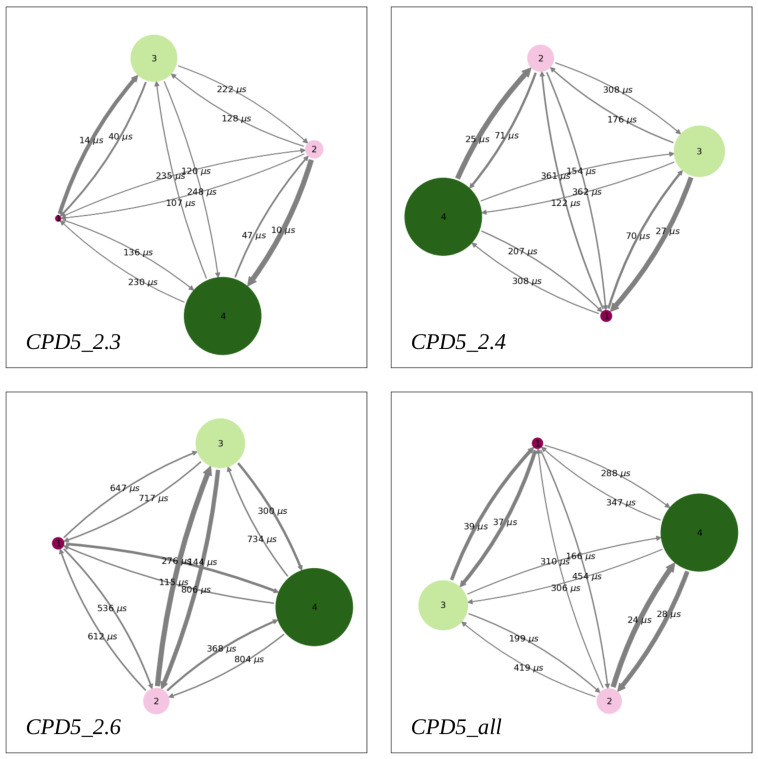
PyEMMA metastable states are represented by colored discs with a radius proportional to their population. The most populated states (labeled 3 and 4) always identify the water phase, while the smaller ones (labeled 1 and 2) represent the two lipid leaflets. Transitions are described with arrows whose thickness and label are associated to the rate constant and MFPT, respectively.

**Figure 6 ijms-24-05021-f006:**
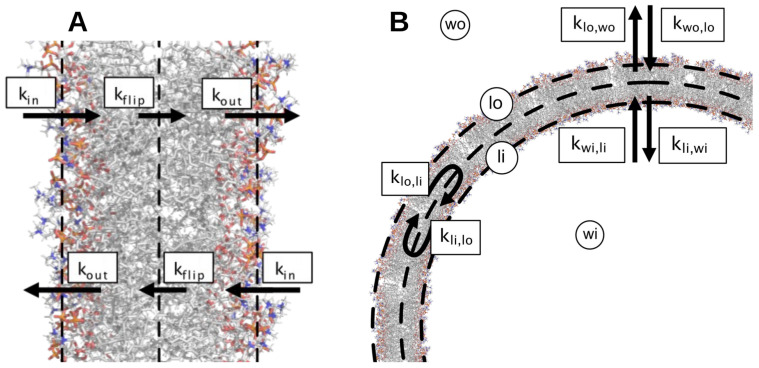
Designation of the kinetic rate constants in membrane planar (**A**) and spherical (**B**) geometry. In the liposome model, wo and lo stands for the outer water and lipid compartment, respectively, and similarly for the two inner compartments wi and li.

**Table 1 ijms-24-05021-t001:** Membrane permeability coefficient results for the cyclic decapeptide CDP5. PAMPA value was taken from [25].

Method	$\log\left[P\left(cm/s\right)\right]\pm \sigma_{P}$
PAMPA	−6.7
ISDM	−8.1 ± 0.4
MSM CDP5_2.3	−1.0 ± 0.1
MSM CDP5_2.4	−1.2 ± 0.1
MSM CDP5_2.6	−1.4 ± 0.1
MSM CDP5_all	−1.0 ± 0.1

**Table 2 ijms-24-05021-t002:** Markov State Models built from four different sets of 100 ns unconstrained trajectories of the peptide-membrane system.

MSM Name	Nb of Trajectories	Initial Positions
CDP5_2.3	2 × 75	*z* = 0.0 and 2.3 nm
CDP5_2.4	2 × 75	*z* = 0.0 and 2.4 nm
CDP5_2.6	2 × 75	*z* = 0.0 and 2.6 nm
CDP5_all	4 × 75	*z* = 0.0, 2.3, 2.4, and 2.6 nm

## Data Availability

All molecular dynamics trajectories supporting results reported in this study are available on request from the corresponding author. The raw data are not publicly available due to their size which prevent them from being uploaded to a publicly available repository.

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
