# Peer review of "Understanding Passive Membrane Permeation of Peptides: Physical Models and Sampling Methods Compared"

_ijms, 2023, doi:10.3390/ijms24055021_

Round 1

Reviewer 1 Report

The manuscript of Mazzanti & Ha-Duong deals with the calculation of the permeability coefficient of molecules/peptdides traversing a membrane bilayer. Overcoming cellular barriers is a key process for molecules that need to reach internal targets, and sometimes it represents a bottleneck in the design of new drugs because of the difficulties to quantify it.

In this work the authors applied two different methodologies based on molecular simulations to compare the prediction of the permeability coefficient of a peptide for which experimental data are available. Eventually they reported also the computational cost of the two methods.

The manuscript is well written and presented. Simulations were performed with state-of-the-art techniques employing a well-known software package. It has a quite broad interest since the comparison between the two methodologies is useful for many topics where rare events occurs, not only in the case of diffusion through a membrane.

The paper merits to be published, I have a just a few small remarks:

-       In then introduction they speak implicitly about the entry of peptides/molecules in human cells. However, the same problem of permeation occurs for Gram negative bacteria, where two different membranes (as composition) protect the interior. For the outer membrane some porins or active systems can help the diffusion Thus molecules favored to enter in the periplasmic space might have problems to reach the cytoplasm. The authors might add some references because the same methods can be applied there. (see for example https://doi.org/10.1038/s41579-019-0294-2 for passive systems and https://doi.org/10.1021/acsinfecdis.2c00202 for active systems).

-       Introduction pag 2: they refer to membrane permeation occurring in the second-minute time scale. It would be interesting to note that with the barrier they found using the ISDM, 12 kcal/mol, the Transition State Theory would predict a passage in the ms range. Does the difference with the prediction of Table 2 come from the (inhomogeneous) diffusion coefficient?

-       Table 2, the units [cm/s] probably do not refer to log(P), but to P only, it might be misleading as written now.

-       Pag. 16 first paragraph: “…the permeability though a planar membrane. ” is probably to be intended as “…the permeability through a planar membrane. ”

Reviewer 2 Report

In the present study, authors have done MD simulations study of CDP5 with POPC membrane. Although the topic is good, the authors have performed quite a good analysis. But there is a technical issue with the parameterization of CDP5, authors have used the charmm-general force field (CGFF) to generate parameters for this peptide, which is incorrect Since the CGFF force field is for small drug-like molecules not for Amino acids peptide, however, there are parameters for some nonstandard amino acids are present in CGFF. But in the case of cyclic peptide partial charge distribution would be different on amino acid residue atoms. To correctly parameterize the peptide author either could use residue-specific ff (https://pubs.acs.org/doi/abs/10.1021/acs.jcim.9b00647), or obtain the correct charges using QM calculations. 

Round 2

Reviewer 2 Report

The authors have addressed my concern. I would recommend acceptance of the manuscript.